# High-Performance Reversible Furan–Maleimide Resins Based on Furfuryl Glycidyl Ether and Bismaleimides

**DOI:** 10.3390/polym15163470

**Published:** 2023-08-19

**Authors:** Jiawen Wang, Jixian Li, Jun Zhang, Shuyue Liu, Liqiang Wan, Zuozhen Liu, Farong Huang

**Affiliations:** 1Key Laboratory for Specially Functional Materials and Related Technology of Ministry of Education, School of Materials Science and Engineering, East China University of Science and Technology, Shanghai 200237, China; wjw563230507@163.com (J.W.); ilsyue@163.com (S.L.);; 2Huachang Polymers Co., Ltd., East China University of Science and Technology, Shanghai 200241, China

**Keywords:** Diels–Alder reaction, furan–maleimide resin, thermally reversible cross-linking resins, high-performance resin

## Abstract

Two reversible furan–maleimide resins, in which there are rigid -Ph-CH_2_-Ph- structures and flexible -(CH_2_)_6_- structures in bismaleimides, were synthesized from furfuryl glycidyl ethers (FGE), 4,4′-diaminodiphenyl ether (ODA), *N*,*N*’-4,4′-diphenylmethane-bismaleimide (DBMI), and *N,N′*-hexamethylene-bismaleimide (HBMI). The structures of the resins were confirmed using Fourier transform infrared analysis, and the thermoreversibility was evidenced using differential scanning calorimetry (DSC) analysis, as well as the sol-gel transformation process. Mechanical properties and recyclability of the resins were preliminarily evaluated using the flexural test. The results show the Diels–Alder (DA) reaction occurs at about 90 °C and the reversible DA reaction occurs at 130–140 °C for the furan–maleimide resin. Thermally reversible furan–maleimide resins have high mechanical properties. The flexural strength of cured FGE-ODA-HBMI resin arrives at 141 MPa. The resins have a repair efficiency of over 75%. After being hot-pressed three times, two resins display flexural strength higher than 80 MPa.

## 1. Introduction

Thermosetting resins are a class of highly cross-linking polymers that are often mechanically strong and hard [1]. Therefore, thermosetting resins are indispensable in the fields of composite materials, structural adhesives, electronic packaging materials, and protective coatings [2]. Thermosetting resins form three-dimensional networks connected to each other by covalent bonds after curing. Covalent bonds are usually strong but irreversible, making thermosetting resins difficult to be reprocessed. This runs counter to the goal of achieving a circular economy [3]. Fortunately, introducing dynamic reversible bonds allows for the recycling of strong thermosetting resins [1]. Moreover, dynamic reversible reactions produce no additional products and do not demand any other reactants, meaning they are carbon-economical. Nowadays, dynamic covalent bonds are widely used to prepare dynamically reversible cross-linked resins. The reactions that generate dynamic covalent bonds include Diels–Alder (DA) reactions [4,5], disulfide bond exchange reactions [6,7], interesterification reactions [8,9], transamidation reactions [10], imine and acylhydrazone bond exchange reactions [11,12], siloxane equilibrium reactions [13], alkyl transfer reactions [14], and olefin metathesis reactions [15].

The DA reaction is ideal for preparing thermally stimulated repair polymeric materials due to its good thermal reversibility, high yield, few side reactions, and mild reaction conditions. Most studies have focused on the DA reaction between furan (diene) and maleimide (dienophile) due to the high reactivity exhibited by the imide group [16]. The furan ring shows a distinct diene character rather than an aromatic ring, which means that its chemical behavior is more similar to cyclopentadiene than benzene [17]. Bismaleimide (BMI) resins are bifunctional thermosetting resins with maleimide as the reactive end group [18]. BMI resins have high mechanical properties and dimensional stability. However, they are not recycled due to the high crosslinking structures after curing [19]. Furan–maleimide resins based on the DA reaction undergo a [4 + 2] reaction between furan rings and maleimide groups at lower temperatures. The reversible-DA (r-DA) reaction occurs above 120 °C so that furan–maleimide resins are depolymerized into furan and maleimide [20]. Fan et al. [21] prepared a series of recyclable polymers via a Diels–Alder reaction between furan-functionalized poly(hydroxyamino ether) and 1,5-bis(maleimide)-2-methylpentane. In their work, the crosslink density of the recyclable network was adjusted by varying the content of the flexible-(CH_2_CH_2_O)-structure. They found that with the rise in the content of the flexible structure, the glass transition temperature decreased while the r-DA reaction temperature increased. Currently, most of the furan–maleimide resins reported in the literature have flexural strengths below 50 MPa [22,23]. The introduction of flexible chains is a common method to improve the mechanical properties of resins [24]. Ma et al. increased the impact strength by up to 100% by introducing the poly(ethylene glycol) structure into polytriazole resins [25].

In this paper, two reversible furan–maleimide resins with either a rigid -Ph-CH_2_-Ph- structure or a flexible-(CH_2_)_6_- structure in the bismaleimide units were prepared by one-pot synthesis using furfuryl glycidyl ether (FGE), 4,4′-diaminodiphenyl ether (ODA), *N,N′*-4,4′-diphenylmethane-bismaleimide(DBMI), and *N,N′*-hexamethylene-bismaleimide (HBMI) as the raw materials. The curing behaviors, thermal properties, mechanical properties, and reversible properties of the resins were investigated and, meanwhile, the effect of flexible structures on the properties was discussed.

## 2. Materials and Methods

### 2.1. Materials

Furfuryl alcohol, tetrabutylammonium bromide, epichlorohydrin, maleic anhydride, sodium hydroxide (NaOH), anhydrous sodium sulphate (Na_2_SO_4_), ethyl acetate, 1,6-hexanediamine, *N,N′*-dimethylformamide (DMF), sodium acetate (AcONa), acetic anhydride ((Ac)_2_O), triethylamine (TEA), sodium bicarbonate (NaHCO_3_), petroleum ether, 4,4′-oxydianiline (ODA), and *N,N′*-4,4′-diphenylmethane-bismaleimide (DBMI) were obtained from Shanghai Titan Technology Co., Shanghai, China. All reagents were analytical reagent grade.

### 2.2. Instruments and Measurements

The Fourier transform infrared (FT-IR) spectrum was measured using a Nicolet iS10 infrared spectrometer (Thermo Scientific Corporation, Madison, WI, USA). The scanning range was 400–4000 cm^−1^ with a resolution of 4 cm^−1^ and a scanning time of 32. The proton nuclear magnetic resonance (^1^H-NMR) spectrum was carried out using an AVANCE III 400 spectrometer (Bruker, Billerica, MA, USA) operating at 400 MHz with deuterated chloroform (CDCl_3_) as the solvent and tetramethylsilane (TMS) as the internal standard. The melting point (*T*_m_) was measured using an X4 melting apparatus (Shanghai Analytic Instrument Factory, Shanghai, China). The electron impact mass spectrometry (EI-MS) spectrum was performed using the GCT Premier EI-TOF mass spectrometer (Waters, Milford, Massachusetts, USA) with an *m*/*z* range of 10–1500 Da. Differential scanning calorimetry (DSC) analysis was carried out with a TA Instruments Q2000 analyzer (TA, New Castle, DE, USA) by setting a ramp from 40 °C to 300 °C with a heating rate of 10 °C/min and a N_2_ flow of 50 mL/min. Thermogravimetric analysis (TGA) was carried out on a TGA/DSC 1LF analyzer (METTLER TOLEDO, Greifensee, Switzerland) in N_2_ with a gas flow rate of 60 mL/min and a heating rate of 10 °C/min. The flexural properties were tested according to GB/T 2567-2021 on an electronic universal testing machine (SANS CMT 4204, Shanghai, China). The three-point flexural model was used for the test, and the test loading speed was 2 mm/min. The size of the sample for the flexural test was 80 × 15 × 4 mm^3^.

The content of soluble components of the resins in acetone with different structures was measured using a Soxhlet extractor. Resins with mass *m*_0_ were put into the extractor and continuous extraction at the reflux temperature of acetone for 24 h was conducted. Then, the resins after extraction were taken out and dried under vacuum. Afterwards, the resins were weighed and recorded as *m*_1_. The content of soluble components (*S*) in the resins was calculated according to Equation (1):(1)$$S=\frac{m_{0}\;-m_{1}}{m_{0}}\text{×}100\%$$

The reversibility of the resins was verified using the hot-pressing process. The resins were first cut into small scraps; then, the scraps were pressed in a flat vulcanizing machine at 140 °C under the pressure of 3 MPa for 30 min. Afterwards, the pressed scraps were continually placed at 90 °C for 24 h.

### 2.3. Synthesis of Furfuryl Glycidyl Ether (FGE)

The synthesis of FGE was shown in Figure 1. Furfuryl alcohol (FAL, 9.82 g, 0.100 mol) and tetrabutylammonium bromide (TBAB, 0.450 g, 14.0 mmol) were added to a three-necked flask equipped with a reflux condenser, a thermometer, and a mechanical stirrer. Epichlorohydrin (ECH 10.2 g, 0.110 mol) was dropped into the flask under inert gas protection and the reaction mixture was stirred vigorously at room temperature for 4 h. Then, the mixture was cooled to 10 °C with an ice bath and 50 wt% NaOH aqueous solution (20 mL) was slowly added into the flask. After the reaction mixture was stirred for 2 h at ambient temperature, 30 mL deionized water was added. The product was extracted with 10 mL ethyl acetate and the organic phase was separated in a separator. The extraction was conducted three times. The organic extracted product was collected and washed three times with deionized water. The organic phase was separated out, dried with anhydrous Na_2_SO_4_, and filtrated to give a transparent solution. Then, the solvent in solution was removed under reduced pressure to get an orange-yellow liquid. In addition, the liquid was further purified using column chromatography with a mixed solvent of acetate/petroleum = 1/1 to obtain a light yellow liquid product (FGE, yield 78%) [16,26]. ^1^H-NMR (400 MHz, CDCl_3_) δ: 7.44–7.39 (t, 1H, -O-C**H**=CH-), 6.38–6.31 (d, 2H, =C**H**-C**H**=), 4.53 (q, 2H, C-C**H_2_**-O), 3.80–3.40 (q, 2H, -O-C**H_2_**-CH), 3.16 (m, 1H, 
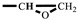
), 2.80–2.61 (t, 2H, 
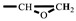
). FT-IR (KBr, cm^−1^): 3146, 3125(=C-H, υ), 3062 (-C-H on the oxirane ring, υ), 1504 (-C=C-, υ), 1257, 850 (C-O-C on the oxirane ring, υ), 1217, 1084 (CH_2_-O-CH_2_, υ), 1150 (C-O-C on the furan ring, υ), 1008 (C-O-C on the furan ring, δ), 918 (C-O-C on the oxirane ring, δ), and 751 (single-substituted furan ring, δ). EI-MS (*m*/*z*): 154 [M]^+^, 97 [M-
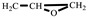
]^+^, 81 [M-
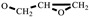
]^+^ (Appendix A).

### 2.4. Synthesis of N,N′-Hexamethylene-Bismaleimide (HBMI)

The synthesis of HBMI is shown in Figure 2. 1,6-Hexanediamine (HDA, 11.6 g, 0.100 mol) and DMF (100 mL) were added to a four-necked flask equipped with a reflux condenser, a thermometer, and a mechanical stirrer. Maleic anhydride (21.6 g, 0.220 mol) dissolved in DMF (100 mL) was dropped into the flask and the mixture was stirred at 90 °C for 3 h. At the end of the reaction, the mixture was cooled to 60 °C, after which AcONa (2.00 g, 0.0244 mol) was added and a mixture of (Ac)_2_O (100 mL) and TEA (20 mL) was dropped into the flask. Then, the mixture was heated at 90 °C for another 3 h. The reaction solution was washed with ice water. The solid was separated and washed with saturated NaHCO_3_ followed by washing with deionized water several times to obtain a yellow powder. Finally, the powder was dried in a vacuum oven at 100 °C for 12 h. A light yellow powder product was obtained, *N,N′*-hexamethylene-bismaleimide (HBMI, yield 56%, *T*_m_ 137–138 °C) [27]. ^1^H-NMR (400 MHz, CDCl_3_) δ: 6.60 (s, 4H, **H**C=C**H**), 3.39 (t, 4H, -NC**H_2_**-), 1.56–1.34 (m, 4H, -NCH_2_C**H_2_**-), 1.34–1.09 (m, 4H, -NCH_2_CH_2_C**H_2_**). FT-IR (KBr, cm^−1^): 3107 (=C-H, υ), 2942 (-CH_2_-, υ), 1690 (C=O, υ), 1587, 1453 (C=C, υ), 1227 (C-N, υ), and 698 (=C-H, δ). EI-MS (*m*/*z*): 276 [M]^+^, 110 [M-
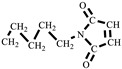
]^+^, 82 [M-
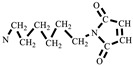
]^+^ (Appendix A).

### 2.5. Synthesis of Cured FGE-ODA-BMI Resins

Prior to the preparation of resins, a preliminary investigation of the reaction temperatures between the different monomers was carried out, as detailed in the Appendix A. The synthesis of two FGE-ODA-BMI resins, namely FGE-ODA-DBMI resin and FGE-ODA-HBMI resin, was shown in Figure 3. The synthesis of FGE-ODA-DBMI was taken as an example to demonstrate the synthesis details. FGE (5.15 g, 33.0 mmol), *N,N′*-4,4′-diphenylmethane-bismaleimide (DBMI 5.98 g, 16.7 mmol), and DMF (5 mL) were added to a 100 mL three-necked flask equipped with a stirrer, a thermometer, and a condenser. After that, the temperature of the mixture was slowly raised to 55 °C under stirring until the mixture became transparent. Then, 4,4′-diaminodiphenyl ether (ODA, 1.65 g, 8.25 mmol) was added into the flask. The mixture was stirred well to ensure homogenous mixing. A certain amount of the mixture was put into a preheated mold covering a release agent in a 125 °C vacuum oven for 30 min to remove the solvent. Then, a reddish-brown liquid was formed at 125 °C (FGE-ODA-DBMI resin) [28]. After the resin was heated at 110 °C for 12 h and at 90 °C for 24 h, a reddish-brown cured FGE-ODA-DBMI resin was obtained. The cured FGE-ODA-HBMI resin was made in the same way except using HBMI instead of DBMI.

## 3. Results

### 3.1. Curing Procedures and Structural Characterization of FGE-ODA-BMI Resins

The reactions between FGE and DBMI, FGE and ODA, DBMI and ODA, and among DBMI, FGE and ODA are investigated using DSC analysis and the results are shown in Figure 1. As shown in Figure 1a, there is an exothermic peak (1st) in the range of 68–104 °C, an endothermic peak in the range of 124–155 °C, and another exothermic peak (2nd) in the range of 155–205 °C for the mixture of FGE and DBMI. The former exothermic peak (1st) is due to the DA polymerization reaction, and the latter exothermic peak (2nd) is the self-polymerization of DBMI, while the endothermic peak is due to the reversible DA reaction [29,30,31]. There is an exothermic peak between 105 °C and 232 °C in the DSC curve for the mixture of FGE and ODA, which is due to the ring-opening reaction of oxirane rings with amino groups. For the mixture of ODA and DBMI, there is an endothermic peak between 90 °C and 150 °C due to the melting of the mixture. As shown in the curve of the DBMI and ODA, the exothermic phenomenon occurs above 150 °C. Usually, the Michael addition reaction between the -C=C- of DBMI and the amino group of ODA is exothermic and usually occurs above 130 °C [29,30]. Therefore, the exothermic phenomenon is due to the Michael addition reaction between DBMI and ODA. As shown in Figure 1b, the thermal curing reaction of FGE-ODA-DBMI resin includes three stages in the range of 40–160 °C. The first one (67–102 °C) is owing to the DA reaction and the second one (107–125 °C) is due to the ring-opening reaction between oxirane rings and amino groups, and, then, the third one (125–153 °C) is attributed to the r-DA reaction [29,30]. The reactions between FGE and HBMI, FGE and ODA, HBMI and ODA, and among HBMI, FGE and ODA are investigated using DSC analysis and the results are shown in Appendix A.

Three curing procedures, i.e., 90 °C/36 h, 110 °C/36 h, and 110 °C/12 h + 90 °C/24 h, for the resins are designed. The FGE-ODA-DBMI resins are cured under three different curing procedures, and the resultant cured resins are noted as cured FGE-ODA-DBMI-1 resin, cured FGE-ODA-DBMI-2 resin, and cured FGE-ODA-DBMI-3 resin, respectively. The flexural properties of cured FGE-ODA-DBMI resins are shown in Figure 2a. Cured FGE-ODA-DBMI-3 resin has the highest flexural strength (125 MPa) and flexural modulus (4.30 GPa). For cured FGE-ODA-DBMI-1 resin, the ring-opening reaction between oxirane rings and amino groups has not yet taken place at 90 °C. Thereby, cured FGE-ODA-DBMI-1 resin has the lowest flexural strength. When the resin is heated at 110 °C for 36 h, the ring-opening reaction between oxirane rings and amino groups partially occurs while the DA reaction is not obvious during this curing temperature, meaning the resulting resin is further cross-linked. The flexural properties further increase. When the resin is heated at 110 °C for 12 h and, then, at 90 °C for another 24 h, the partially cross-linked resin continues to form DA bonds, leading to a higher cross-linking degree; hence, cured FGE-ODA-DBMI-3 resin has the highest flexural properties. The effect of three curing procedures on the flexural properties of the cured FGE-ODA-HBMI resin is also investigated and the results are shown in Figure 2b. Similarly, cured FGE-ODA-HBMI-3 resin possesses the highest flexural properties. Therefore, the curing procedure (110 °C/12 h + 90 °C/24 h) is optimal for FGE-ODA-DBMI and FGE-ODA-HBMI resins. The FGE-ODA-BMI resins will be cured with the above optimal curing procedure unless otherwise stated.

FT-IR spectroscopy is used to analyze the changes in the characteristic functional groups of the cured FGE-ODA-DBMI resin and its raw material, as shown in Figure 3a. The disappearance of the asymmetrical stretching (918 cm^−1^) of oxirane rings in the cured FGE-ODA-DBMI resin and the appearance of the extensive absorption at about 3450 cm^−1^ corresponding to hydroxyl groups generated through the ring opening reaction between ODA and FGE reveal that oxirane rings react with amino groups [29,30]. The peaks at 2930 cm^−1^ and 2860 cm^−1^ are attributed to the antisymmetric and symmetric stretching vibration of -CH_2_ in turn, and the peak at 1424 cm^−1^ is due to the bending vibration of -CH_2_ [29,30]. The appearance of a new peak at 1776 cm^−1^ corresponding to succinimide rings generated through the DA reaction confirms the successful synthesis of cross-linked networks. The pattern of change for the cured FGE-ODA-HBMI resin is similar and shown in Appendix A. Meanwhile, FT-IR spectroscopy is used to analyze the changes in the characteristic functional groups of FGE-ODA-DBMI resin during the curing process (Figure 3b). As the curing time of FGE-ODA-DBMI resin increases, the C-O-C stretching vibration peak on the oxirane ring at 1254 cm^−1^ gradually decreases, and the C-N characteristic peak at 1668 cm^−1^ gradually increases. When the resin is cured at 110 °C for 12 h, the C-O-C peak disappears, and the C-N peak no longer increases. This result shows that oxirane rings react with amino groups. A new peak is identified at 1776 cm^−1^, which is specific to the DA adduct of maleimide [25]. The absorption peak at 1776 cm^−1^ no longer increases when the curing procedure changes from 110 °C/12 h + 90 °C/24 h to 110 °C/12 h + 90 °C/30 h, indicating that the curing procedure (110 °C/12 h + 90 °C/24 h) for FGE-ODA-DBMI resin is reasonable. The pattern of change for FGE-ODA-HBMI resin is similar and would not be repeated (Appendix A). A Soxhlet extractor is used to measure the content of soluble components in resins of different structures to determine the curing degree of the resin. The content of soluble components in cured FGE-ODA-DBMI resin and cured FGE-ODA-HBMI resin are 2.21% and 1.29%, respectively. The content of soluble components decreases when the flexible structure is introduced into the resin (FGE-ODA-HBMI). This demonstrates that cured FGE-ODA-HBMI resin possesses a higher cross-linking degree than cured FGE-ODA-DBMI resin.

### 3.2. Thermal Property of the Cured FGE-ODA-BMI Resins

The thermoreversibility of the resins is characterized using DSC analysis and the DSC curves for cured FGE-ODA-BMI resins are shown in Figure 4. As shown in Figure 4a, the reversible Diels-Alder (r-DA) reaction peak temperature of the cured FGE-ODA-HBMI resin (141 °C) is higher than that of the cured FGE-ODA-DBMI resin (130 °C). This is because the flexible structure in the cured FGE-ODA-HBMI resin gives molecular chains greater mobility. The TGA curves of the resins with different structures are shown in Figure 4b. The decomposition does not happen below 150 °C, which further ensures the thermoreversibility. There are two possible reasons for the weight loss of the resin at 150–300 °C. One reason is that the resins undergo the r-DA reaction below 155 °C, depolymerizing into FGE-ODA and BMI (shown in Appendix A). Based on the TGA curves of FGE-ODA and BMI, it can be seen that FGE-ODA shows significant weight loss below 300 °C (as shown in Appendix A). Therefore, the weight loss of the resin below 300 °C is attributed to the weight loss of FGE-ODA. Another possible reason is that the DA adducts dehydrate to form a benzene ring [32]. 

### 3.3. Reversible Performance of the Cured FGE-ODA-BMI Resins

The reversibility of the cured FGE-ODA-BMI resin is further analyzed using DSC tests. As shown in Figure 5a, the cured FGM-ODA-DBMI is firstly heated from 20 °C to 140 °C at a heating rate of 10 °C·min^−1^ (1st heating) and immediately cooled to 20 °C at a cooling rate of 10 °C·min^−1^ (1st cooling), and, then, heated to 140 °C again at the same heating rate (2nd heating). There is an endothermic peak at 130 °C and 131 °C in 1st heating and 2nd heating, respectively. At the same time, an exothermic peak appears at about 90 °C in 1st cooling. The results indicate that for the cured FGE-ODA-DBMI resin, the DA reaction occurs at about 90 °C and the r-DA reaction at about 130 °C. As shown in Figure 5b, an endothermic peak for the cured FGE-ODA-HBMI resin appears at 142 °C and 143 °C in the 1st heating and 2nd heating, respectively. Meanwhile, there is an exothermic peak at about 95 °C in 1st cooling. The results indicate that for the cured FGE-ODA-HBMI resin, the DA reaction occurs at about 95 °C, and the r-DA reaction does at about 140 °C. 

The reversibility of cured FGE-ODA-BMI resins is verified using the sol-gel transformation process. The cured FGE-ODA-DBMI resin is taken as an example. The cured FGE-ODA-DBMI resin slightly swells after being immersed in DMF at 50 °C for 6 h. The phenomenon indicates that the cured FGE-ODA-DBMI resin is cross-linked and thus insoluble in DMF (Figure 6b). Then, when heated at 140 °C, the resin quickly turns into a reddish-brown liquid (Figure 6c), which indicates that the r-DA reaction of oxanorbornenes in the network has taken place. Due to the reduction in molecular weights, the depolymerized resin is easily dissolved in DMF. Subsequently, when the liquid is heated at 90 °C for a period of time, the broken DA bonds reunite to form a cross-linked network. As the DA reaction progresses, the cross-linking degree gradually increases. After 4 h, the resin forms a gel (Figure 6d). Finally, the resin repolymerizes into a reddish-brown lumpy solid. The sol-gel transformation process of the cured FGE-ODA-HBMI resin is similar to that of the cured FGE-ODA-DBMI resin (Appendix A). These results show that the cured FGE-ODA-DBMI resin and the cured FGE-ODA-HBMI resin are thermosetting resins with good thermal reversibility. 

Since the resin is thermally reversible, the fractured resin could be reshaped using the hot-pressing method to form a homogeneous solid (Appendix A). The schematic repair process of the cured FGE-ODA-BMI resins is shown in Figure 7.

The repair efficiency of the resin is determined by testing the flexural strength of the resin after hot pressing. As listed in Table 1, cured FGE-ODA-DBMI and FGE-ODA-HBMI resins possess the flexural strength of 125 MPa and 141 MPa, and the flexural moduli of 4.30 GPa and 4.25 GPa, respectively, indicating that the flexible structures in the resin could increase flexural strength but slightly reduce flexural modulus. Because cured FGE-ODA-DBMI resin has a lower r-DA reaction peak temperature, the r-DA reaction is easier to occur, which is beneficial to repair. Thus, the repair efficiency of cured FGE-ODA-DBMI resin is higher than that of cured FGE-ODA-HBMI resin. In addition, the flexural property of the resin declines after repair. This is because furan rings generated by the depolymerization of DA bonds are easily oxidized at high temperatures [33].

After the cured FGE-ODA-BMI resins were reshaped using the hot-pressing method, the flexural properties are tested and the retention rate is calculated. The initial resins are named R-0, and the resins reprocessed for the first, twice, and third time are named R-1, R-2, and R-3, respectively. As shown in Table 1, after the cured FGE-ODA-DBMI resin is recompressed three times, the flexural strength is 83 MPa, giving a retention rate of 66.4% compared to the initial flexural strength. Similarly, the cured FGE-ODA-HBMI resin has a retention rate of 61.1% after being repressed three times.

## 4. Conclusions

In this paper, two kinds of thermally reversible cross-linking resins (FGE-ODA-DBMI, FGE-ODA-HBMI) are prepared by changing the structure of BMI. In addition, the properties of FGE-ODA-DBMI and FGE-ODA-HBMI resins are studied. First, the curing process of the two resins is studied using DSC and FT-IR analyses. The optimal curing process of 110 °C/12 h + 90 °C/24 h is determined. Secondly, the reversibility of the cured FGE-ODA-DBMI and FGE-ODA-HBMI resins is analyzed using DSC tests. The DA reaction occurs at about 90 °C and the r-DA reaction occurs at about 140 °C for FGE-ODA-BMI resins. At the same time, the thermal reversible behavior of resin is verified using the sol-gel transformation process. Finally, the cured resin is reshaped using the hot-pressing method to form a homogeneous solid. The cured FGE-ODA-DBMI and FGE-ODA-HBMI resins have a repair efficiency of over 75% at the beginning. After being repressed three times, both resins retain more than 60% of flexural strength. Compared to the FGE-ODA-HBMI resin, the FGE-ODA-DBMI resin with a rigid structure has a lower flexural property but better reversibility. These results would provide a reference for the design and preparation of high-performance reversible resins.

## Data Availability

The datasets generated during and/or analyzed during the current study are available from the corresponding author on reasonable request.

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
