# Peer review of "High-Performance Reversible Furan–Maleimide Resins Based on Furfuryl Glycidyl Ether and Bismaleimides"

_polymers, 2023, doi:10.3390/polym15163470_

Round 1

Reviewer 1 Report

The authors present the “High performance reversible furan-maleimide resins based on furfuryl glycidyl ether and bismaleimides”. However, it should be revised before it can be published.

1)      Lines 174-181 & Lines 185-187: references have to be added for all of these reactions.

2)      Lines 182-183: What exothermic phenomena do the authors refer to?

3)      Lines 187-189: The reactions between FGE and HBMI, FGE and ODA, HBMI and ODA, and among HBMI, FGE and ODA in Figure S3 have to be discussed. what do they add to the specific manuscript?

4)      Lines 193-194: Why did the authors select the specific curing procedures?

5)      It is necessary to add references for the functional groups that the FT-IR spectrum examined.

6)      Why is the pattern of change for FGE-ODA-HBMI resin the same?

7)      How did the authors calculate the reversible Diels-Alder (r-DA) reaction peak temperature of the cured FGE-ODA-HBMI resin in Figure 4?

8)      The inset figure in Figure 4b does not have a good resolution.

9)      The cured FGE-ODA- HBMI resin seems to be dehydrated in the range of 150-300 °C.

10)  Figure 6: How did the authors calculate the peaks observed? A peak fitting analysis has to be used to calculate the peaks observed.

11)  Lines 278-279: “The results indicate that for the cured FGE-ODA-HBMI resin, the DA reaction occurs at about 95°C, and the r-DA reaction does at about 140°C.” Why this happens?

12)  “Rretention rate” in Table 1 has to be corrected in the revised manuscript.

13)  Τhe English language and the structure of the manuscript (spaces, alignment, etc.) should be carefully considered.

Τhe English language and the structure of the manuscript (spaces, alignment, etc.) should be carefully considered.

Reviewer 2 Report

This manuscript presents the synthesis of dynamic epoxy resins using furfuryl glycidyl ether (FGE), diamine, and bismaleimides (DBMI or HBMI). The reversible thermosetting resins obtained were characterized through DSC analysis and FT-IR spectroscopy. The thermal properties of these resins were also investigated. The work presented is interesting and has potential for publication after revision. However, there are several points that need to be addressed:

1) The presented methodology for synthesizing the epoxy resins involves a two-step process. First, FGE and bis-maleimide are heated at 55 °C in DMF. Then, ODA is added and the mixture is further heated at 125 °C. It is important to provide information on the degree of conversion of bis-maleimide in the reaction with FGE. If the conversion is not complete, the contribution of side reactions, such as Michael addition between bis-maleimide and ODA, should be evaluated.

 2) The authors need to provide further evidence for the thermal aromatization of oxanorbornene fragments in the FGE-ODA-BMI resin. Aromatization of oxanorbornenes typically requires strong acidic conditions. Therefore, the authors should exclude the possibility of side processes, such as decomposition of the furan rings or bis-maleimide, contributing to the observed weight loss. Additionally, an experiment on the aromatization of the FGE/ODA adduct should be conducted and included.

3) It is necessary to provide the glass transition temperature and/or TGA data for the dynamic epoxy resins.

4) SEM micrographs of the repaired resins after hot-pressing should be included to visualize their repairing ability.

5) The NMR spectra are not correctly presented. Integration values and peak labels are missing.

6) On page 4, line 155, please amend to "… FGE (5.15 g, 33.0 mmol)…"

7) Please separate the temperature degrees with a space from the number.

Round 2

Reviewer 1 Report

Τhe work has improved quite a bit. However, the peak fitting analysis that is used to calculate the peaks observed is not correct. Did the authors use a program to fit the curves? There is no subtraction from the baseline. The selected boundaries of the curves for the fitting analysis are not correct.

Minor editing of English language required

Reviewer 2 Report

In the revised version of the manuscript, the authors have addressed most of my questions except for the first comment. I have noticed some confusion in the experimental section of the manuscript and in response to this comment. Based on the information provided in the experimental part (section 2.5), the synthesis of resins was carried out using the following methodology: " … FGE (5.15 g, 33.0 mmol), N,N'-4,4'-diphenylmethane-bismaleimide (DBMI 5.98 g, 16.7 mmol), and DMF (5 mL) were added to a 100 mL three-necked flask equipped with a stirrer, a thermometer and a condenser. After that, the temperature of the mixture was slowly raised to 55 °C under stirring until the mixture became transparent. Then 4,4′-diaminodiphenyl ether (ODA, 1.65 g, 8.25 mmol) was added into the flask. The mixture was stirred well. A certain amount of the mixture was put into a preheated mold covering a release agent in a 125 °C vacuum oven for 30 min to remove residual solvents." The initial reaction between FGE and DBMI should occur at 55 °C, rather than a ring-opening reaction between FGE and ODA as mentioned in reply. Therefore, if the reaction system contains ODA, BMI and FGE (assuming the initial reaction between FGE and BMI was not complete), there is a possibility of a non-reversible Michael addition between BMI and ODA, as well as a reaction between FGE and ODA, which will compete with each other after heating at 125 °C. Therefore, to exclude the non-desired Michael addition process, the authors should provide the selectivity of the initial reaction between FGE and BMI under the conditions used (55 °C, DMF). To ensure a clearer understanding of the synthesis process of cured FGE-ODA-BMI resins, the authors should provide a more detailed description of the experiments, including the addition of reaction time and the selectivity of transformations occurring at each stage.

Round 3

Reviewer 1 Report

Accept in present form. The manuscript has been checked to see if it follows the instructions of the journal.

Minor editing of English language required

Author Response

Dear reviewer, thanks for your warm advice. Based on your suggestions, we have made reasonable modifications on the grammar of the article. Your suggestions helped a lot to improve the readability of the article.

Reviewer 2 Report

The authors have not conducted additional experimental studies to verify the structure of the resulting polymers. Based on the DSC data, it is evident that reactions between FGE and ODA, as well as between ODA and DBMI, are competing processes in the mixture of FGE, ODA, and DBMI at temperatures around 120-130 °C. The authors suggest that FGE and DBMI may not undergo the DA reaction at 55 °C, and ODA did not undergo the Michael addition reaction with DBMI at 125 °C. However, these claims should be experimentally confirmed. To do so, the authors should provide the results of the reaction between FGE and ODA in deuterated DMF at 55 °C, as well as the reaction between ODA and DBMI in deuterated DMF at 125 °C, followed by NMR analysis. Without this essential data, I cannot recommend this work for publication.

Round 4

Reviewer 2 Report

I thank the authors for conducting the necessary experiments that confirmed the previously stated claims about the structure of the obtained polymers. Taking into account the obtained experimental data, I recommend this article for publication